# Tissue- and Population-Level Microbiome Analysis of the Wasp Spider *Argiope bruennichi* Identified a Novel Dominant Bacterial Symbiont

**DOI:** 10.3390/microorganisms8010008

**Published:** 2019-12-19

**Authors:** Monica M. Sheffer, Gabriele Uhl, Stefan Prost, Tillmann Lueders, Tim Urich, Mia M. Bengtsson

**Affiliations:** 1Zoological Institute and Museum, University of Greifswald, 17489 Greifswald, Germany; gabriele.uhl@uni-greifswald.de; 2LOEWE-Center for Translational Biodiversity Genomics, Senckenberg Museum, 60325 Frankfurt, Germany; stefanprost.research@protonmail.com; 3South African National Biodiversity Institute, National Zoological Gardens of South Africa, Pretoria 0001, South Africa; 4Bayreuth Center of Ecology and Environmental Research, University of Bayreuth, 95448 Bayreuth, Germany; tillmann.lueders@uni-bayreuth.de; 5Institute of Microbiology, University of Greifswald, 174897 Greifswald, Germany; tim.urich@uni-greifswald.de

**Keywords:** microbiome, symbiosis, endosymbiont, transmission, range expansion, Araneae, spiders, *Argiope bruennichi*, invertebrate host, Tenericutes

## Abstract

Many ecological and evolutionary processes in animals depend upon microbial symbioses. In spiders, the role of the microbiome in these processes remains mostly unknown. We compared the microbiome between populations, individuals, and tissue types of a range-expanding spider, using 16S rRNA gene sequencing. Our study is one of the first to go beyond targeting known endosymbionts in spiders and characterizes the total microbiome across different body compartments (leg, prosoma, hemolymph, book lungs, ovaries, silk glands, midgut, and fecal pellets). Overall, the microbiome differed significantly between populations and individuals, but not between tissue types. The microbiome of the wasp spider *Argiope bruennichi* features a novel dominant bacterial symbiont, which is abundant in every tissue type in spiders from geographically distinct populations and that is also present in offspring. The novel symbiont is affiliated with the Tenericutes, but has low sequence identity (<85%) to all previously named taxa, suggesting that the novel symbiont represents a new bacterial clade. Its presence in offspring implies that it is vertically transmitted. Our results shed light on the processes that shape microbiome differentiation in this species and raise several questions about the implications of the novel dominant bacterial symbiont on the biology and evolution of its host.

## 1. Introduction

All multicellular life evolved from and with microbes. Consequently, the interactions between animals and microbes are not rare occurrences, but rather fundamentally important aspects of animal biology from development to systems ecology [1]. The holobiont, defined as a host and all of its symbionts, is considered as a unit of biological organization, upon which selection can act [2,3,4,5]. The nature of the relationships between host and symbionts has been of intense interest in recent years; while some form obligatory, coevolutionary symbioses [6,7,8,9,10], others are environmentally derived and/or unstable and temporary [11,12]. The collective microbial symbionts and their environment within a certain host or tissue can also be referred to as a microbiome [13]. For example, the intensive research on the human microbiome of the last decade has shed light on many roles of the microbiome of different tissues in health and disease [14]. In addition, correlations have been found between the microbiome and a number of traits across different levels of biological organization and states (from population-level [15] down to the level of tissue-specific microbiomes [14,16] as well as across different age and disease states [17]).

A striking feature of the microbiomes of some hosts is the presence of microbial endosymbionts. Endosymbionts, which typically reside within the cells of their hosts, can play a major role in speciation in many organisms through mechanisms such as assortative mating and reproductive isolation [18]. *Wolbachia* endosymbiont infections are highly prevalent in invertebrates [19,20] where they can induce parthenogenesis, cause cytoplasmic incompatibility between uninfected and infected individuals, affect host fecundity, fertility, and longevity [21,22], and affect the sex ratio of host species via feminization of males and male killing [23,24,25]. Non-*Wolbachia* (endo)symbiotic bacteria can also manipulate host physiology and behavior in diverse ways, from increasing heat tolerance in aphids [26] to determining egg-laying site preference in *Drosophila melanogaster* [27]. If microbial symbionts are vertically transmitted, these modifications in behavior and/or physiology can result in changing selection pressures and eventually the coevolution of the symbionts and their hosts [4,6,28,29,30].

The function of a symbiont within its host is often predictive of its location within tissues. *Wolbachia* infections are often specifically located in reproductive tissues, but can also be distributed widely throughout somatic cells, depending on the host species [31,32]. Beyond *Wolbachia*, many studies on bacterial symbionts have focused on blood- and sap-feeding insects; these specialist feeders require symbionts within their digestive tissues to assist in the utilization of their nutrient-poor diets [6,33,34,35,36,37,38,39,40]. Therefore, endosymbiont presence, and thus microbiome composition, can vary widely between tissue types and organ systems.

Among arthropods, insects have been the primary focus of microbiome studies. In comparison, investigations into the microbiome of spiders are scarce but suggest that spiders host diverse assemblages of bacteria, some of which alter their physiology and behavior. In a survey of 8 spider species from 6 different families, in which DNA (deoxyribonucleic acid) was extracted from the whole body, putative endosymbionts dominated the microbiome of all species [41]. The endosymbionts discovered (assumed by the authors to be endosymbionts of the spiders, not endosymbionts of their insect prey) were largely reproductive parasites, including *Wolbachia, Cardinium, Rickettsia, Spiroplasma,* and *Rickettsiella,* which corresponds to the findings on other spider species across families [42,43,44]. The non-endosymbiont bacterial taxa were typical insect gut microbes, which could be nutritional symbionts of the spiders or represent the microbiome of prey the spiders consumed. As to the effect of endosymbionts on spider hosts, relatively little is known. *Wolbachia* has been shown to bias the sex ratio in the dwarf spider *Oedothorax gibbosus* [45] and *Rickettsia* infection changed the dispersal probability of another dwarf spider species *Erigone atra* [46]. The abundance of *Rhabdochlamydia* was found to vary with population and with sex (higher infection rate in females than males) in *Oedothorax gibbosus* [44]. The studies mentioned above have focused on endosymbionts alone within a single family of spiders. It has not yet been investigated whether there are intraspecific differences in the total (endosymbiont and non-endosymbiont) microbial community between different spider populations, the composition of the microbiome in certain tissue types, or whether there is vertical transmission of the microbiome in spiders.

*Argiope bruennichi* (Scopoli, 1772), an orb-weaving spider with a Palearctic distribution [47], is an ideal candidate for a pioneering microbiome study given the wealth of knowledge that exists on the biology of the species and the genus *Argiope* [48]. It has been the subject of many studies due to a number of interesting traits, such as sexual dimorphism and sexual cannibalism (i.e., [49,50,51]), and its recent and rapid range expansion within Europe [47,52,53,54,55]. Since spider dispersal behavior can also be affected by endosymbiont infection [46] and dispersal behavior influences the rate of range expansion, the microbiome might play a role in the rapid range expansion of *A. bruennichi*. Although some studies on *A. bruennichi* have used targeted approaches to look for specific reproductive parasites, finding none [43,56], a holistic approach to investigating the microbiome of *A. bruennichi* has not been carried out to date. In the present study, we investigated the total bacterial community of *A. bruennichi* from geographically distant, but genetically similar, populations in Germany and Estonia, asking the following questions: (1) does *A. bruennichi* possess a multi-species microbiome? (2) If so, are there population-level differences in the microbiome? (3) Are specific microbes localized in certain tissues? And (4) is the microbiome vertically transmitted? 

## 2. Materials and Methods 

### 2.1. Sample Collection

For this study, mature female *Argiope bruennichi* were collected for two purposes: first, for dissection into different tissue types and, second, to produce offspring. The females used for dissection came from two sites: one in Germany (Greifswald: 54.11 N, 13.48 E; *n* = 3) and one in Estonia (Pärnu: 58.30 N, 24.60 E; *n* = 3). These locations were selected because they are geographically distant from one another and yet are in the same haplotype group, according to a previous population genetic study [54], thereby controlling for evolutionary differentiation of the microbiome within the species. The sample size (*n* = 3 per sampling site) was decided upon as the minimum size required for statistical tests of variability within and between collecting sites due to the time-intensive and delicate nature of dissection. The females that produced offspring came from two sites (Plech, Germany: 49.65 N, 11.47 E; *n* = 1; Pärnu, Estonia: 58.30 N, 24.60 E; *n* = 1) and were maintained in the lab until they produced an egg sac. It is important to note that *A. bruennichi* females lay their eggs into a simple egg sac, which is then wrapped in a silk casing consisting of two layers: one “fluffy” silk layer and one tough outer layer [57]. Eggs hatch within the first weeks, but the juvenile spiders, “spiderlings,” remain in the egg sac for several months over winter [57]. The spiderlings that hatched from the egg sacs produced in the lab were preserved in the silk casing in the freezer until the day of DNA extraction for microbiome analysis.

### 2.2. Sample Preparation

Three adult specimens each from Greifswald and Pärnu were dissected within two days of collection and the spiders were not fed between the point of collection and dissection. Before dissection, the spiders were anaesthetized using CO_2_, after which the prosoma and opisthosoma were separated using sterilized scissors. A 10 μL sample of hemolymph was immediately taken from the aorta at the point of separation with a sterile pipette. Next, the legs were removed and a single leg was taken as a sample and stored separately from the whole prosoma. Sterilized forceps were used for dissection of the opisthosoma. The cuticle was removed dorsally and a sample of the midgut was taken from the dorsal side and stored. The cuticle was then cut ventrally, between the epigynum (genital opening) and the spinnerets. The two cuticular flaps were pulled to loosen the internal organs and the digestive tubules were teased apart to reveal the rest of the organs. The major ampullate silk glands, which produce structural and dragline silk and are the largest and easiest to remove of all the silk glands [58,59,60,61], were removed and stored. Then, a sample of the ovaries was removed and stored. Removal of the ovaries revealed the cloaca and existing fecal pellets and the surrounding fluid in the cloaca were sampled using a sterile pipette. Finally, the book lungs were removed and stored. All tissue samples were stored in sterile tubes and frozen until the time of DNA extraction. 

For the spiderling samples, one egg sac each from Plech and Pärnu was opened with sterilized forceps and 5 spiderlings from each egg sac were placed directly into phenol-chloroform for DNA extraction.

### 2.3. DNA Extraction and Illumina Amplicon Sequencing

DNA was extracted from tissue samples using a phenol-chloroform extraction protocol, as described in [62]. Mechanical lysis was performed via bead beating in a FastPrep 24 5G (MP Biomedicals, Irvine, CA, USA) with FastPrep Lysing Matrix E. A fragment of the 16S rRNA gene was amplified from the extracted DNA with a primer pair recommended by the Earth Microbiome Project, targeting the V4 region of the 16S rRNA gene (515f: 50-GTGYCAGCMGCCGCGGTAA-30, 806r: 50-GGACTACNVGGGTWTCTAAT-30 [63]) coupled to custom adaptor-barcode constructs. PCR amplification and Illumina MiSeq library preparation and sequencing (V3 chemistry) was carried out by LGC Genomics in Berlin. Sequences have been submitted to the NCBI short read archive and can be found under the BioProject number PRJNA577547, accession numbers SAMN13028533- SAMN13028590. 

In addition, PacBio long-read SMRT (single molecule real-time) sequencing of almost full-length 16S rRNA gene amplicons was performed for two of the samples (a prosoma extract from a German spider and a spiderling extract from Estonian spiderlings). For this, ~1500 bp amplicons were amplified using the primers Ba27f (AGAGTTTGATCMTGGCTCAG) and Ba1492r (CGGYTACCTTGTTACGACTT) tailed with PacBio universal sequencing adapters (universal tags) in a first round of PCR with 25 cycles. After PCR product purification, a second round of PCR was done with distinct barcoded universal F/R primers as provided by the manufacturer (PacBio, Menlo Park, CA, USA). SMRTbell Library preparation and SMRT sequencing on a PacBio Sequel System was also done according to manufacturer instructions. Approximately 20 barcoded amplicons were multiplexed per SMRT cell. Initial processing of SMRT reads and exporting of CCS (circular consensus sequencing) data was done with the SMRT Link analysis software as recommended by the manufacturer. Raw reads are available on the NCBI short read archive and can be found under the BioProject number PRJNA577547, accession number SAMN13046638. 

The resulting sequences were clustered and consensus sequences derived using IsoCon [64]. The highest abundant sequence, dubbed DUSA (dominant unknown symbiont of *Argiope bruennichi*–see Results section) was identified by comparing the short V4 amplicon with the SMRT IsoCon consensus sequences and choosing the sequence with the highest match.

### 2.4. Sequence Processing

Sequences clipped from adaptor and primer sequence remains were received from the LGC Genomics sequencing facility and then processed using the DADA2 (divisive amplicon denoising algorithm 2) package in R (Version 1.6.0 [65]) [66]. The R script used for sequence processing can be found in Appendix A. Forward and reverse Illumina reads were simultaneously filtered and truncated to 200 bp. Error rates were estimated using the maximum possible error estimate from the data as a first guess. Sample sequences were de-multiplexed and unique sequences were inferred using the core denoising algorithm in the DADA2 R package. Following sample inference, paired forward and reverse reads were merged. Chimeric sequences accounted for less than 0.5% of the total sequence reads and were removed using the removeBimeraDenovo function. Taxonomic classification was performed using the DADA2 package’s implementation of the RDP’s naïve Bayesian classifier [67], with a minimum bootstrap confidence of 50, drawing from the Silva database [68]. The resulting unique amplicon sequence variants (ASVs) with taxonomic classification were used to build a table containing relative abundances of ASVs across all samples. 

### 2.5. Data Analysis and Visualization

To control for possible contamination during the process of extraction and sequencing, given low DNA yield from some tissue types, a control extraction using sterile water was performed alongside each extraction. These negative controls were included in the sequencing run. A series of cutoffs were employed as quality control on the relative abundance table. First, samples with low sequencing depth (less than 4000 reads) were removed. Then, the data was strictly filtered to remove any ASVs found in extraction blanks (with an abundance of 50 reads or more). After the removal of those possible contaminants, another sequencing depth cutoff was enforced, removing samples with less than 400 reads.

ASVs were aggregated by bacterial class to obtain an overview of the microbiome. Low-abundance classes (less than 1000 reads total, meaning less than 0.1% of filtered reads) were aggregated into a category called “Other.” The relative abundance of each class was then visualized in the form of pie charts using the ggplot2 package [69] in R. 

To test for and visualize dissimilarity in ASV composition between tissue types, sampling sites, and individuals, non-metric multidimensional scaling was performed on Hellinger-transformed sequence variant counts using Bray–Curtis distance, implemented in the vegan package (vegan function ‘metaMDS’) (version 2.5-1 [70]) in R. Hellinger transformation was used to account for differences in library size and to reduce the effect of low abundance sequences. Explanatory power of tissue type, sampling site, and individual was calculated using a PERMANOVA test (vegan function ‘adonis’). This analysis was done on filtered reads, once with the most dominant ASV (DUSA) excluded due to its overwhelming influence on the data, which might mask the patterns of the rest of the bacterial community, and once with DUSA included. The R script used for filtering, statistical analysis, and data visualization of the 16S amplicon sequences can be found in Appendix A.

The almost-full length 16S rRNA gene sequence of DUSA generated by SMRT amplicon sequencing was compared to that of well-known endosymbiotic bacterial taxa retrieved from Silva and GenBank, along with two archaeal sequences as an outgroup. The sequences were aligned using ClustalW implemented in MEGA [71,72] and a consensus tree was calculated using IQ-TREE [73] with 5000 bootstrap iterations. The consensus tree was visualized using FigTree [74]. For clarity of visualization, branches were collapsed by phylum for distant taxa and by genus for Tenericutes; for an un-collapsed tree of the Tenericutes and all accession numbers see Appendix A and Appendix A. 

## 3. Results

Illumina amplicon sequencing of the V4 region of the 16S SSU rRNA (small subunit ribosomal ribonucleic acid) gene of 6 adult spiders (8 tissue types each) and two spiderling samples from 2 locations resulted in 5.2 million reads with an arithmetic mean of 90,377 reads per sample (min = 711, max = 981,405). Of total raw reads, 86.8% passed quality filtering and chimera removal. Chimeras counted for less than 0.5% of all reads. We first removed samples with low sequencing depth (less than 4000 reads), which eliminated 4 samples: 1 prosoma sample from Estonia, 1 silk gland sample from Estonia, 1 hemolymph sample from Estonia, and 1 ovary sample from Estonia. We then removed sequences with high abundance in negative controls (more than 50 reads in control samples), resulting in the removal of a total of 337 possible contaminant sequences. Lastly, we again filtered out samples with low sequencing depth (less than 400 reads), eliminating 4 more samples: 1 leg sample from Estonia, 1 hemolymph sample from Estonia, 1 fecal pellet sample from Estonia, and 1 hemolymph sample from Germany. After these filtering steps, 1.77 million reads remained, with an average of 41,182 reads per sample (min = 477, max = 629,137). In total, post-filtering, 574 amplicon sequence variants (ASVs) were detected in the tissues and spider populations. 

### 3.1. A Bacterial Symbiont in *Argiope bruennichi*

The microbiome of *A. bruennichi* was dominated by a single ASV, making up 84.56% of all filtered reads (Figure 1). This ASV had less than 85% identity to any sequence in the NCBI (National Center for Biotechnology Information) database. Long read sequencing of two samples generated a near full length 16S rRNA gene amplicon sequence corresponding to the dominant ASV, which allowed us to further investigate the identity of this dominant symbiont (Table 1). All low-similarity matches originated from environmental samples and uncultured microbes. There was no match to a named taxon, making it difficult to classify the sequence taxonomically. An exploratory gene tree (Figure 2) placed the sequence within the Tenericutes, which are gram-negative, cell-associated bacteria that have lost their cell walls [75]. We refer to this dominant unknown symbiont as DUSA (dominant unknown symbiont of *Argiope bruennichi*) henceforth. 

After filtering, 573 additional ASVs were detected in the samples, the majority of which were assigned to seven bacterial classes: Actinobacteria (75 ASVs), Alphaproteobacteria (96 ASVs), Bacilli (60 ASVs), Bacteroidia (49 ASVs), Clostridia (84 ASVs), Gammaproteobacteria (115 ASVs), and Mollicutes (3 ASVs). Details of the ASVs in these most abundant classes can be found in Appendix A. ASVs with the highest abundance (more than 500 reads post-filtering), other than DUSA, were identified as the genera *Mesoplasma* (Mollicutes: Entomoplasmatales: Entomoplasmataceae), *Acinetobacter* (Gammaproteobacteria: Pseudomonadales: Moraxellaceae), *Micrococcus* (Actinobacteria: Micrococcales: Micrococcaceae), *Frigoribacterium* (Actinobacteria: Micrococcales: Microbacteriaceae), and *Alcaligenes* (Gammaproteobacteria: Betaproteobacteriales: Burkholderiaceae). Archaea were not detected. 

### 3.2. Tissue Localization and Population Differentiation

With DUSA excluded from the analysis, tissue types did not differ significantly in microbiome community composition (PERMANOVA, R^2^ = 0.180, *p* = 0.366). However, microbiome community composition varied significantly between populations (PERMANOVA, R^2^ = 0.045, *p* < 0.01) and individuals (PERMANOVA, R^2^ = 0.059, *p* < 0.001). The interaction between individual and population was also significant (PERMANOVA, R^2^ = 0.044, *p* < 0.01) (Figure 3).

With DUSA included in the analysis, the results were similar but *p* and R^2^ values were slightly different: tissue type: PERMANOVA R^2^ = 0.231, *p* = 0.131; population: PERMANOVA R^2^ = 0.039, *p* < 0.1; individual: PERMANOVA R^2^ = 0.040, *p* < 0.1; and interaction of individual and population: PERMANOVA R^2^ = 0.057, *p* < 0.05. 

### 3.3. Vertical Transmission

Juvenile spider (spiderling) samples were dominated by DUSA (Figure 1). Other bacterial classes made up less than 6% of the filtered reads in spiderlings from Germany and less than 0.001% of reads in spiderlings from Estonia.

## 4. Discussion

### 4.1. An Unknown Symbiont Dominates the *Argiope bruennichi* Microbiome

We demonstrated that *A. bruennichi* spiders contain a multi-species microbiome, answering the first of our research questions. However, the *A. bruennichi* microbiome is dominated by an unknown symbiont sequence (DUSA). DUSA likely represents a novel bacterial clade, due to the low sequence identity to known taxa [76]. A robust evolutionary placement is not possible without further genomic analysis. However, our gene tree suggested that it is likely a close relative or member of the Tenericutes. Due to this placement within the Tenericutes, DUSA may have similar attributes to other arthropod-associated symbionts in the phylum. It is important to note that we use the word “symbiont” in the broadest sense of the term, as an intimate relationship between two organisms, whether that relationship be mutualistic, parasitic, etc. (sensu de Bary, 1879). The Mollicutes, a class within Tenericutes, contain a number of species known to be associated with arthropods. These mollicute species are generally endosymbiotic and are vertically transmitted [77,78]. Their effects on hosts are diverse: some are pathogenic [79], while others increase host fitness under parasitism [80], or form nutritional mutualisms via nutrient recycling [78]. In such close symbioses, the endosymbiont genomes usually evolve much faster than free-living species; this can be due to gene loss and/or gain if the hosts provide metabolites to their endosymbionts and vice versa [81,82,83,84,85]. This tendency toward rapid evolution of endosymbionts may explain the low 16S rRNA sequence similarity to other bacteria in the database and would suggest that DUSA forms a close relationship, such as endosymbiosis, with the spider host. 

Of the three mollicute ASVs detected in our samples, two were assigned to the genus *Spiroplasma*, but were detected in very low abundance. The third was assigned to the genus *Mesoplasma* and was the second-most abundant ASV in our study. It was only found to be abundant in German spiders and primarily in midgut and fecal pellet samples from a single individual. If this *Mesoplasma* ASV is a facultative nutritional symbiont of the spider (i.e., [77,78] for *Mesoplasma* in insects), we would expect it to be present in most investigated members of a species or population. Alternatively, it could be a symbiont of the spider prey, which is more likely since *Mesoplasma* and its relatives are very common symbionts of insects [42,77,78,86,87]. Considering that *Mesoplasma* was found only in the midgut and fecal pellets, it can be assumed that it is prey-derived and its presence within the host is transient.

### 4.2. The *Argiope bruennichi* Microbiome Varies between Individuals and Populations, but Not between Tissues

Our analysis of the microbial community composition of tissue types, individuals, and populations showed that there is high variability between all samples. Because the *A. bruennichi* microbiome is dominated by DUSA, the other ASVs had lower sequencing coverage, which could contribute to the observed variability. Alternatively, the sequencing coverage may be representative of a true lack or low abundance of other microorganisms if DUSA has a high fitness within the host and thereby outcompetes other bacteria. Despite this, we found significant differences between individuals and between populations, thereby answering our second research question. It could be that the microbiome (excluding DUSA) of these spiders is transient and taken up from the environment and especially from their diet, as is the case in some insects [11]. For instance, across many butterfly species, the larval microbiome largely reflects the microbiome of the food plant’s leaves [12]. To test the hypothesis of a partly prey-derived microbiome for *A. bruennichi*, future studies could sequence both the microbial and prey communities by combining the methods used in our study with gut content sequencing, as described in [88]. Different prey communities between populations and individuals (at the time of sampling) could lead to the differences observed in our study.

We found no significant differences in the microbial community between tissue types, with or without DUSA included in the analysis, addressing our third research question. Although endosymbiont infections are often localized within reproductive tissues, which could lead to tissue differentiation [31,32], infection of somatic tissues may facilitate horizontal transfer of a symbiont through feces, as in the Triatomid bug vectors of Chagas disease [89], or to parasites, as in the case of a *Nasonia* wasp and its fly host [90]. There are also cases of symbionts that live primarily in insect hemolymph and are thus found in all tissues [91,92]. Tissue differentiation could also arise in the presence of nutritional symbionts in the gut of a host, but no study has explicitly tested this in spiders. Additionally, there are no reported cases of nutritional symbionts in spiders. If there are differences between organ systems in *A. bruennichi*, they are too subtle be detected with the current sample size.

### 4.3. Evidence of Vertical Transmission of DUSA

We analyzed the microbiome of spiderlings to address our fourth research question, whether the microbiome of *A. bruennichi* is vertically transmitted. Our data suggest that at least DUSA is indeed vertically transmitted. Spiderling samples contained a high abundance of DUSA reads and few other ASVs. Spiderlings could recruit bacteria from the environment or from their mothers via different avenues. Environmental colonization could possibly occur before or after the closing of the silken egg sac, in the moments between oviposition and encasement in silk, or by passing through the tough outer case (refer to the Methods section for a description of *A. bruennichi* egg sac components). We consider these environmental avenues to be unlikely given the extremely short amount of time that the eggs are exposed to the environment before encasement (M.M. Sheffer, G. Uhl, personal observation) and because *A. bruennichi* egg sac silk is extremely dense and egg sac silk of other spider species has been shown to inhibit growth of bacteria [93]. Vertical transmission of bacteria from mother to offspring could occur while the eggs are in the ovaries or by deposition during the egg-laying process. We consider vertical transmission to be the most likely avenue for bacterial presence within spiderling tissue, supported by the low diversity of bacteria found in spiderling samples and the presence of DUSA in female ovaries. Whether transmission occurs before or after egg laying could be tested using fluorescence in situ hybridization to visualize DUSA in or on eggs. Taken together, the high divergence of DUSA from other bacterial taxa and its evident vertical mode of transmission suggest the potential for a tight coevolutionary relationship between DUSA and *A. bruennichi.*

### 4.4. Implications for Future Studies of *Argiope bruennichi* and Beyond

The presence of an endosymbiont might explain the incongruence between mitochondrial and nuclear DNA markers found by a study investigating the phylogeographic history of *A. bruennichi* [47]. The authors offered three possible explanations for this result: male-biased dispersal, selection on mitochondria, or reproductive parasites (e.g., *Wolbachia* spp.). The authors considered the last explanation the least likely as no previous study had identified *Wolbachia* spp. or other reproductive parasites in *A. bruennichi* [42,47,56]. However, these studies targeted a handful of known reproductive parasites using specific primers and PCR (polymerase chain reaction) assays [42,56], which excluded the possibility of discovering any novel symbionts. Given our discovery of DUSA, the hypothesis that infection with reproductive parasites caused incongruence between molecular markers in *A. bruennichi* should be revisited. To that end, future efforts should focus on characterizing DUSA, for example, by in-depth genomic analysis to determine its phylogenetic placement as well as by exploring its distribution across the host species’ range and its localization and functions inside the host. Further investigations could illuminate whether the relationship between *A. bruennichi* and DUSA is pathogenic, commensal, or mutualistic. Importantly, the presence and/or absence of DUSA in other spider or insect species should be explored, perhaps thereby providing clues into the origin of this novel symbiosis.

Our study adds to a growing body of literature suggesting that bacterial symbionts, especially endosymbionts, play an important role in spider biology. Two other recent studies that surveyed the microbiomes of several spider species found putative endosymbiotic taxa to be both prevalent (70% of surveyed individuals [94]) and dominant within certain hosts (>90% of bacterial reads [41,95]). We demonstrated, in addition, that spiders are a source of novel symbiont taxa, which make them interesting targets for discoveries of new types of symbiotic interactions that may impact host biology in yet unimaginable ways. Several unique aspects of spider biology make them particularly exciting for studying symbiosis. For example, their predatory lifestyle offers ample opportunities for symbiont taxa from their prey to enter the spider host, in some cases giving rise to new stable associations. In addition, spiders employ external digestion by secreting digestive fluids into their prey, which sets them apart from the internal digestive systems of most insect hosts that have until now been the subject of (endo)symbiosis research. The implications of these peculiarities on spider-bacterial interactions are yet unchartered territory, opening up promising new research avenues on symbiosis.

## 5. Conclusions

Our study is the first to look into the localization of microbial symbionts in spider tissues. The principal discovery was that of a novel symbiont, which was found to dominate the microbiome of all individuals and tissue types investigated. Its characteristics, such as low sequence identity to other bacteria and possible vertical transmission, suggest that it may belong to a novel clade of bacterial endosymbionts with a tight association to its host. Our findings highlight the need for holistic microbiome studies across many organisms, which will increase our knowledge of the diversity and evolution of symbiotic relationships.

## Figures and Tables

**Figure 1 microorganisms-08-00008-f001:**
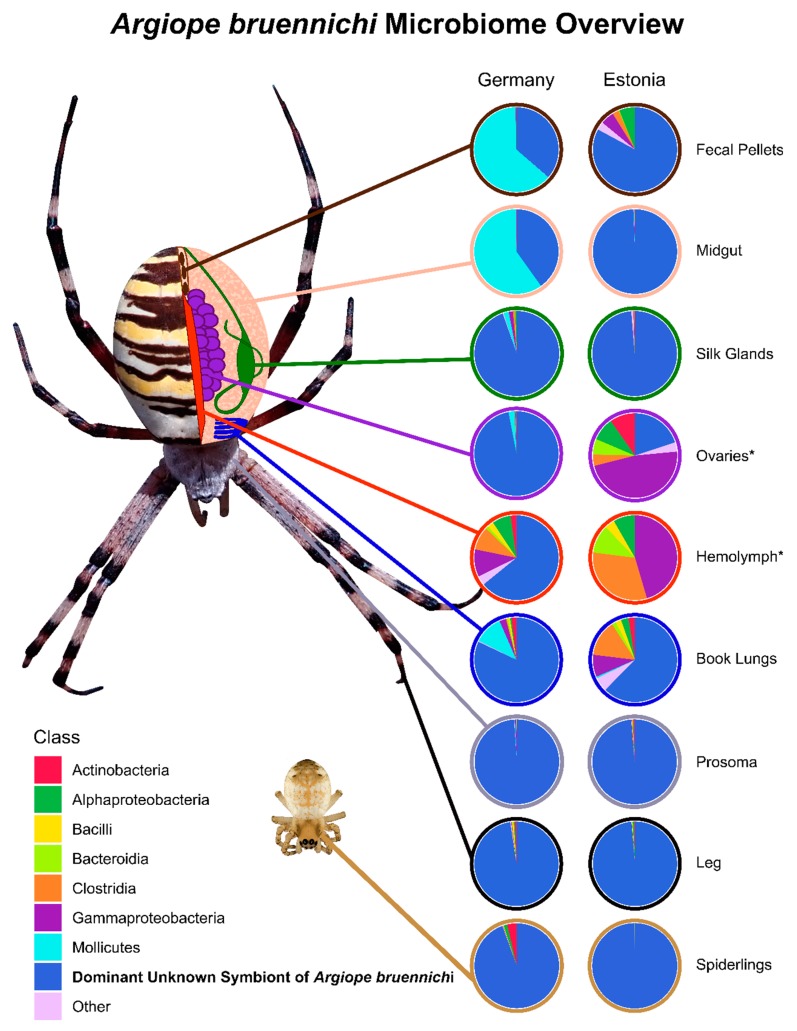
Microbiome composition of spider tissue types and spiderlings from Germany and Estonia. Tissue types are represented in a schematic drawing of *Argiope bruennichi* internal anatomy. 16S rRNA gene sequences were pooled by class. Classes with low abundance were combined into an “Other” category. The dominant unknown symbiont of *Argiope bruennichi* (DUSA) is separated from other unknown sequences, which were of low abundance. Asterisks (*) denote tissue types that had a sample size lower than two (Estonia Ovaries: *n* = 1, Estonia Hemolymph: *n* = 1) due to problems with extraction.

**Figure 2 microorganisms-08-00008-f002:**
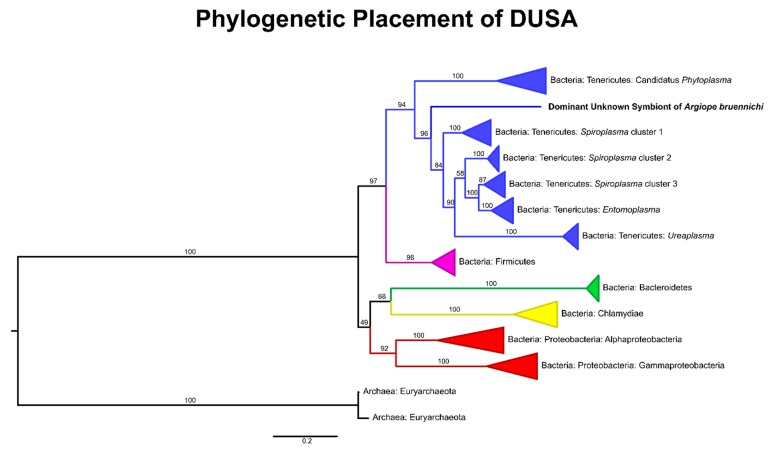
Gene tree placing DUSA relative to endosymbiotic taxa, based on alignment of 16S rRNA gene sequences obtained from Silva and GenBank. Branch labels represent bootstrap support; branches were collapsed by phylum for taxa distantly related to DUSA and by genus for taxa within the Tenericutes. For all accession numbers see Appendix A and for an un-collapsed tree of the Tenericutes see Appendix A.

**Figure 3 microorganisms-08-00008-f003:**
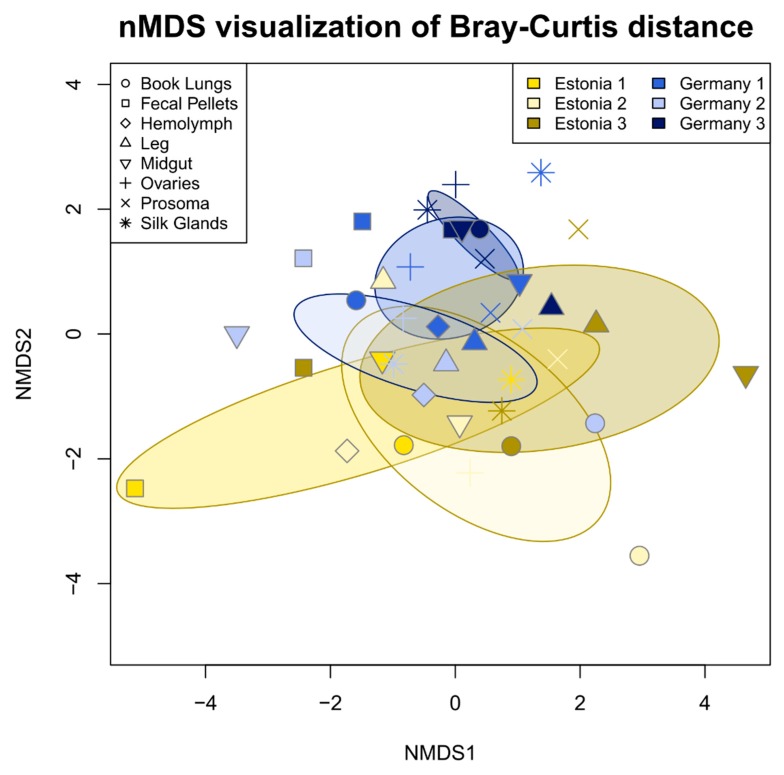
nMDS ordination based on Bray-Curtis distance of 16S rRNA gene sequence variant relative abundance (excluding DUSA) revealed the slight, but significant, differentiation of the *Argiope bruennichi* bacterial community composition according to population (Estonia or Germany in the legend) and individual (denoted by number in the legend) as well as the interaction between the two. Single points represent sequenced tissue samples and the shape of the point represents the tissue type. Shared color denotes tissue samples taken from a single individual spider. Shades of yellow represent spiders collected from Estonia, while shades of blue represent spiders collected from Germany. Ellipses represent the 99% confidence interval based on standard error.

**Table 1 microorganisms-08-00008-t001:** Best matches of the dominant unknown symbiont of *Argiope bruennichi* (DUSA) short and long amplicons in different databases. Results from BLASTN searches against GenBank and from SILVA ACT analysis, as of October 2019.

Query Sequence	GenBank NR Best Match: Taxonomy (Accession number): Sequence Identity %	GenBank Bacteria & Archaea Best Match: Taxonomy (Accession Number): Sequence Identity %	Silva SSU 138 NR: Phylum; Class; Order; Family: Sequence Identity %
ASV V4 region (248bp)	Uncultured prokaryote clone Otu01661 (MG853790.1): 84.3%	*Holdemania filiformis* strain J1-31B-1 (NR_029335.1): 79.92%	Firmicutes; Erysipelotrichia; Erysipelotrichales; Erysipelotrichaceae: 78.7%
Near full-length 16S gene (1492bp)	*Mycoplasma* sp. (e.g., *Biomphalaria glabrata*) (CP013128.1): 82.3%	*Spiroplasma eriocheiris* CCTCC M 207170 strain CRAB (NR_125517.1): 80.79%	Tenericutes; Mollicutes; Entomoplasmatales; Spiroplasmataceae: 79.2%

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
