# Peer review of "Tissue- and Population-Level Microbiome Analysis of the Wasp Spider Argiope bruennichi Identified a Novel Dominant Bacterial Symbiont"

_microorganisms, 2019, doi:10.3390/microorganisms8010008_

Round 1

Reviewer 1 Report

The manuscript entitled “Tissue- and population-level microbiome analysis of the wasp spider Argiope bruennichi identifies a novel dominant bacterial symbiont” presents the microbiome analysis of the spider Argiope bruennichi from similar populations in Germany and Estonia.

 The manuscript presents very interesting results and it is very well written and organized, but in order to be published in Microorganisms journal, it should be completed and/or modified as detailed below:

Q1.The authors are advised to explain what was the purpose of collecting the spider Argiope bruennichi from two sites (Germany and Estonia)?

Q2.The authors are advised to explain why they used the presented sample size. The authors are advised to explain first the abbreviations before using them (ex. DUSA – page 4)

Q3.The authors should rephrase the following: „For now, the implications of these peculiarities for symbiotic interactions between spiders and bacteria is unchartered territory, opening up promising new research avenues on symbiosis”(lines 375-377)

Q4.The authors should rephrase the following: „Although inference is limited by sample size, our findings highlight the need for more holistic microbiome studies across many organisms, which will  increase our knowledge of the diversity and evolution of symbiotic relationships”(lines 383-385)

Author Response

Author’s reply to the Review Report (Reviewer 1)

We thank Reviewer 1 for the input on and improvement of our manuscript, and have answered their questions here and amended the manuscript to address them.

Q1. The authors are advised to explain what was the purpose of collecting the spider Argiope bruennichi from two sites (Germany and Estonia)?

These locations were selected because they are geographically distant from one another, and yet are in the same haplotype group, according to a previous population genetic study (Krehenwinkel and Tautz, 2013), thereby controlling for evolutionary differentiation of the microbiome within the species and only allowing for ecological differentiation. This reasoning has now been added to the manuscript in the Sample collection section of Materials and Methods (lines 108-111).

Q2. The authors are advised to explain why they used the presented sample size. The authors are advised to explain first the abbreviations before using them (ex. DUSA – page 4)

The sample size (n = 3 per sampling site) was decided upon as the minimum size required for statistical tests of variability within and between collecting sites, due to the time-intensive and delicate nature of dissection. This information has now been added to the manuscript (lines 111-113).

The first use of DUSA is now accompanied by an explanation of the abbreviation (lines 166-168). We sincerely thank both reviewers for noticing this mistake.

Q3. The authors should rephrase the following: „For now, the implications of these peculiarities for symbiotic interactions between spiders and bacteria is unchartered territory, opening up promising new research avenues on symbiosis” (lines 375-377)

This sentence has been rephrased to: “The implications of these peculiarities on spider-bacterial interactions is yet unchartered territory, opening up promising new research avenues on symbiosis.” (Lines 395-396)

Q4. The authors should rephrase the following: „Although inference is limited by sample size, our findings highlight the need for more holistic microbiome studies across many organisms, which will increase our knowledge of the diversity and evolution of symbiotic relationships” (lines 383-385)

We have rephrased this sentence to, “Our findings highlight the need for holistic microbiome studies across many organisms, which will increase our knowledge of the diversity and evolution of symbiotic relationships.” (Lines 403-405)

Reviewer 2 Report

The present manuscript describes a detailed analysis of the microbiome of the arachnid species Argiope bruennichi.  The authors performed an extensive 16S rRNA gene sequencing on a comprehensive set of organs and tissues, a rather painstaking process which can result in weak sequencing results.  They obtained over 5.2 million reads (line 210), which after removal of low depth sequencing resulted in a dataset of 1.7 million reads, -yet they were able to identify about 572 organisms in addition to the main symbion which was discovered herein.  This alone merits the publication of the results.  In addition the paper contains at least 2 additional contributions to software development (sequencing processing with DADA2 and the Vegan algorithm for Data Analysis).  This is quite impressive considering that other in other "omics" type of manuscripts no such information is provided.  Given the Scope of this journal the manuscript should be considered within the areas of systems microbiology and microbial ecology, though it might be of interest to microbial physiologists as well.

1) I am not an expert in 16S rRNA (or RNA) sequencing, it would have been nice if there was a more detailed explanation of the process of elimination of reads that had poor sample depth and whether some useful information.  All the sequences have been submitted to the relevant databases.  The data appeared to have been properly processed all the way to the annotated ASVs dataset generation.

2) About the data processing, are the authors applying the Hellinger distance parameter to analyze the sequences in order to assess the Bray-Curtis dissimilarity?  Maybe these parameters have a different meaning in the context of a visual representation.  Also the concept of DUSA is introduced in line 227 of the manuscript, yet it is mentioned on page 4 (line 161). It might be useful to use a footnote or reference it earlier.

3) The predominant 84.5% ASV has at best 85% sequence identity with any of the reported sequences at NCBI, but could it be that there could be errors in either the ASV results or the published 16S rRNA sequence?

4) In the Bra-Curtis distance plot (Fig.3) do the three ellipses from Germany separate fully from the three ellipses from samples from Estonia on a 3D plot? The data for the rRNA from the German sample results seem more dispersed than the organ analogs from the Estonian sample.  Is there an explanation for this discrepancy?

5) The spiderlings have very little are quite similar between the two locations (Fig. 1), but only about half the spider organs have quite that much similarity.  How does the overall distribution for the spiders compare between locations, if that comparison can be done with the data available?

6) Given that the dominant organism is DUSA in all the samples, is it fair to conclude that DUSA must be a endosymbiont or a nutritional mutualist and cannot be a parasite?

7)  The questions are addressed in an orderly manner.  However, what is the evidence for strong variability if only two different spiders were deep sequenced? am I missing something from the population statistical analysis?

8) Related to that, the symbiotic dominance of DUSA could it be due to its fitness within the host such that it outcompetes other bacteria? 

9) The authors suggests that the variability of the less abundant bacteria could originate from the diet, and that sequencing both hosts and plant diets can confirm that.  But what other explanation could there be?

10) The data are very much in agreement with the conclusions of low diversity of the microbiome between different organs and of vertical transmission.  At some point the authors suggest that DUSA is a rapidly evolving organism, but I think that would be inconsistent with its tight association with the host A. bruennichi

I agree that this is a very novel area of research that needs further exploration and that has implication not only for human health but also adaptation to changes in the ecosystem of these species.  Finally the word "principle"  does not apply in line 380, but rather it should be "principal" as in main discovery.

Author Response

Author’s reply to the Review Report (Reviewer 2)

We thank Reviewer 2 for the input on and improvement of our manuscript, and have answered their questions here and amended the manuscript to address them.

1) I am not an expert in 16S rRNA (or RNA) sequencing, it would have been nice if there was a more detailed explanation of the process of elimination of reads that had poor sample depth and whether some useful information.  All the sequences have been submitted to the relevant databases.  The data appeared to have been properly processed all the way to the annotated ASVs dataset generation.

We have now included more information on which samples and how many sequences were removed during the process of filtering (lines 221-229).

2) About the data processing, are the authors applying the Hellinger distance parameter to analyze the sequences in order to assess the Bray-Curtis dissimilarity?  Maybe these parameters have a different meaning in the context of a visual representation.  Also the concept of DUSA is introduced in line 227 of the manuscript, yet it is mentioned on page 4 (line 161). It might be useful to use a footnote or reference it earlier.

The Hellinger transformation was applied on the abundance data prior to calculating Bray-Curtis dissimilarity. This is a common transformation used on species abundance data in ecology, to give low weight to species (in this case, ASVs) with low abundance. We have added a sentence to the text (lines 200-201): “Hellinger transformation was used to account for differences in library size and to reduce the effect of low abundance sequences.”

 The first use of DUSA is now accompanied by an explanation of the abbreviation (lines 166-168). We sincerely thank both reviewers for noticing this mistake.

3) The predominant 84.5% ASV has at best 85% sequence identity with any of the reported sequences at NCBI, but could it be that there could be errors in either the ASV results or the published 16S rRNA sequence?

We considered the possibility of a sequencing error quite seriously when we discovered it, which was our motivation to generate a longer 16S sequence with a different sequencing platform. We now consider errors in the ASV sequence to be unlikely, given that the sequence generated with Illumina amplicon sequencing had a 100% match to a section of the full-length 16S sequence which was independently generated with PacBio sequencing. Some published rRNA sequences within the database may certainly contain errors, but we consider it unlikely that all of the sequences of related taxa within the database would contain errors, resulting in no high percentage match.

4) In the Bra-Curtis distance plot (Fig.3) do the three ellipses from Germany separate fully from the three ellipses from samples from Estonia on a 3D plot? The data for the rRNA from the German sample results seem more dispersed than the organ analogs from the Estonian sample.  Is there an explanation for this discrepancy?

The ellipses in the plot are meant as a visual aid to show the distribution in 2-dimensional space; the statistical test (PERMANOVA) is more indicative of the results, and the low R2 value indicates that there is significant overlap of the groups and that they do not fully separate.

Regarding the difference in dispersion of the points, we cannot offer an explanation at this point. The trend could potentially underlie a true ecological feature, but given the abundance of DUSA and respective low coverage of other taxa, we would need a larger sample size to explore the variation more thoroughly.

5) The spiderlings have very little are quite similar between the two locations (Fig. 1), but only about half the spider organs have quite that much similarity.  How does the overall distribution for the spiders compare between locations, if that comparison can be done with the data available?

That comparison was done using the PERMANOVA analysis, and the location (population) has a significant effect on the microbiome (PERMANOVA, R2 = 0.045, p < 0.01), but the organ (tissue type) does not, although the R2 value is higher (PERMANOVA, R2 = 0.180, p = 0.366).  We cannot do any other comparison with the current data.

6) Given that the dominant organism is DUSA in all the samples, is it fair to conclude that DUSA must be a endosymbiont or a nutritional mutualist and cannot be a parasite?

At present, it is not possible to determine the relationship between DUSA and Argiope bruennichi. It may be mutualistic, parasitic, commensal, etc. In order to test this, further experiments are needed. We do not want to suggest a specific relationship in this study and avoided it within the text. It is important to note that we use the word “symbiont” in the broadest sense of the term, as an intimate relationship between two organisms, whether that relationship be mutualistic, parasitic, etc. (sensu de Bary, 1879). We have added this definition to the discussion (lines 299-301).

7)  The questions are addressed in an orderly manner.  However, what is the evidence for strong variability if only two different spiders were deep sequenced? am I missing something from the population statistical analysis?

The strong variability refers to the amplicon sequencing of all of the spiders in the study. The two spiders which we think are referred to here were sequenced on a PacBio sequencer for the sole purpose of getting a longer sequence for DUSA, and were not used for any assessment of variability. The variability can be seen in the nMDS visualization of Bray-Curtis distance (figure 3) where the points are widely dispersed, and individual identity has a significant effect (high variability between individuals). Due to the low sequencing depth of taxa other than DUSA (resulting either from PCR bias or from true out-competition of DUSA vs. other taxa), the variability between samples was high, and no grouping variable has high explanatory power (R2 value from PERMANOVA test).

8) Related to that, the symbiotic dominance of DUSA could it be due to its fitness within the host such that it outcompetes other bacteria?

Thank you for pointing this out. This is certainly a possibility, and has been added to the discussion (lines 325-327).

9) The authors suggests that the variability of the less abundant bacteria could originate from the diet, and that sequencing both hosts and plant diets can confirm that.  But what other explanation could there be?

If the variability is not solely due to sequencing bias (over-representation of DUSA), we are not confident in any other explanations, other than a transient, diet-driven microbiome. If the reviewer has a suggestion as to an explanation that we have failed to consider, we would be happy to know it and discuss it in the text.

10) The data are very much in agreement with the conclusions of low diversity of the microbiome between different organs and of vertical transmission.  At some point the authors suggest that DUSA is a rapidly evolving organism, but I think that would be inconsistent with its tight association with the host A. bruennichi.

The tight association of an (endo)symbiont with its host can lead to rapid evolution, for example through gene loss as certain needs of the microorganism can be provided for through the host metabolism. The evolution of a symbiont can be, and often is, faster than the evolution of its host, especially considering the generation time of bacteria as compared to animals. We provide several citations to support this claim in the text, and have now added a sentence to make our meaning clearer: “In such close symbioses, the endosymbiont genomes usually evolve much faster than free-living species; this can be due to gene loss and/or gain, if hosts provide metabolites to their endosymbionts, and vice versa [81–85].” (lines 306-307).

I agree that this is a very novel area of research that needs further exploration and that has implication not only for human health but also adaptation to changes in the ecosystem of these species.  Finally the word "principle"  does not apply in line 380, but rather it should be "principal" as in main discovery.

We have corrected the use of “principal” in what was line 380, now line 399. Thank you for your thorough review and catching this error.